# Challenges and Prospects for Designer T and NK Cells in Glioblastoma Immunotherapy

**DOI:** 10.3390/cancers13194986

**Published:** 2021-10-05

**Authors:** Victoria Smith Arnesen, Andrea Gras Navarro, Martha Chekenya

**Affiliations:** Department of Biomedicine, University of Bergen, Jonas Lies Vei 91, 5009 Bergen, Norway; Victoria.Arnesen@uib.no (V.S.A.); gras.andrea@gmail.com (A.G.N.)

**Keywords:** glioblastoma, genomic heterogeneity, natural killer cells, T cells, chimeric antigen receptor, CRISPR/Cas9, immunotherapy

## Abstract

**Simple Summary:**

Designer T and NK cells are a modality within immunotherapy that manipulates receptor-ligand interactions to enhance cells of the immune system to destroy cancer more effectively. Patient’s own immune cells are isolated, genetically modified to improve responses against cancers cells, expanded, and subsequently reintroduced into the individual. Several clinical trials to investigate immunotherapy strategies that may garner lasting benefit even for cancer patients with glioblastoma (GBM), a deadly brain cancer with limited treatment options, are underway. Therapeutic potential for gene editing technologies in limiting rejection and adverse reactions to improve treatment responses in preclinical models is now translating into enduring efficacy in GBM patients.

**Abstract:**

Glioblastoma (GBM) is the most prevalent, aggressive primary brain tumour with a dismal prognosis. Treatment at diagnosis has limited efficacy and there is no standardised treatment at recurrence. New, personalised treatment options are under investigation, although challenges persist for heterogenous tumours such as GBM. Gene editing technologies are a game changer, enabling design of novel molecular-immunological treatments to be used in combination with chemoradiation, to achieve long lasting survival benefits for patients. Here, we review the literature on how cutting-edge molecular gene editing technologies can be applied to known and emerging tumour-associated antigens to enhance chimeric antigen receptor T and NK cell therapies for GBM. A tight balance of limiting neurotoxicity, avoiding tumour antigen loss and therapy resistance, while simultaneously promoting long-term persistence of the adoptively transferred cells must be maintained to significantly improve patient survival. We discuss the opportunities and challenges posed by the brain contexture to the administration of the treatments and achieving sustained clinical responses.

## 1. Glioblastoma—The Current Setting

Cancer is a major health and economic issue worldwide as 19.3 million new cases and 10 million cancer-related deaths were recorded in 2020 [1]. The new reported incidence is projected to surpass 28.4 million by 2040 [2]. The incidence of brain and central nervous system (CNS) tumours was 8.9/100,000 in Europe in 2020 [3], wherein glioblastoma (GBM), the most frequent and aggressive neoplasm, accounted for 3–4/100,000 [4,5]. Although the frequency of GBM is lower than for breast, colorectal, lung, and gastric malignancies, the mortality rate is virtually equivalent [6], as a diagnosis of GBM portends inevitable mortality. GBM is characterised by diffuse infiltration of tumour cells throughout the brain parenchyma and meninges resulting in a poorly demarcated mass that is highly angiogenic and variably hypoxic [7]. Standard treatment of GBM consists of maximal safe surgical resection [8,9], and radiation with concomitant and adjuvant temozolomide (TMZ) chemotherapy [10]. Despite this multimodal approach, recurrence is inevitable, and the overall survival in unselected patients is approximately 12 months after diagnosis [11]. Without therapy, GBM patients die within 3 months, and 5-year survival is less than 10% [12,13,14]. In addition, as improved therapies are increasingly controlling peripheral disease, so will mortality from metastatic lesions to the brain be the major cause of cancer-related death, further emphasizing the need for effective therapies for brain malignancies of all types.

Recent innovations in cancer therapy have shifted the paradigm towards more personalised treatment targeting specific features of the individual patient’s tumour. Unlike many cancers that have a well-defined sequence of events leading to disease, GBM typically occurs de novo. Multiple cells acquire somatic mutations that alter major signalling pathways, resulting in a tumour with high intratumoural heterogeneity [15,16,17]. Immunotherapy, a treatment modality that accentuates the ability of cytotoxic immune cells to recognize malignant cells with particular genetic mutations has demonstrated unprecedented efficacy in several solid malignancies [18,19,20]. These include treatments such as antibody mediated immune checkpoint blockade and adoptive cell transfer of so-called “designer” thymocyte (T) and natural killer (NK) cells [21] armed with chimeric antigen receptors (CAR). This review will focus on CAR T and NK cell immunotherapies for GBM, and the opportunities gene editing technologies offer in improving CAR therapy. We discuss challenges for attaining deep sustained clinical responses in the brain contexture.

## 2. T Cell Patrols with a License to Kill

Patrolling leukocytes detect mutated, early transformed cells, deem them sufficiently “non-self” and co-ordinately eliminate them, as predicted by the cancer immunosurveillance concept [22]. As cells undergo oncogenesis, neoantigens are released and captured on major histocompatibility complex (MHC)/ human leukocyte antigen (HLA) of dendritic cells (DCs) that subsequently mature and migrate to central lymphoid organs. Here, the peptide neoantigen on the DCs’ MHC is presented to the awaiting CD4^+^ or CD8^+^ T cell receptor (TCR) complex. Subsequently, binding of the CD28 co-stimulatory receptor to the DCs’ CD80/86 receptor fully activates the cytotoxic T cells which then migrate to infiltrate the tumour and kill the cells by locally releasing perforin and granzymes [23,24]. These lymphocytes successfully eliminate the genetically unstable tumour cells with intrinsically high immunogenicity [25] through a series of successive stages [26]. The T cells also effectively terminate their activation and proliferation as a means of avoiding autoimmunity, resulting in different phenotypes that either further activate Th1 immune responses or suppress via Th2-driven responses. Surface receptors such as cytotoxic T-lymphocyte-associated protein 4 (CTLA-4), programmed cell death protein 1 (PD-1) and nuclear transcription factors attenuate T cell responses, where CTLA-4 competes with CD28 for binding to CD80/86, resulting in inhibitory downstream signalling [27]. PD-1 is an immunoinhibitory receptor that stymies lymphocyte proliferation and cytokine secretion when bound to its membrane-bound or secreted ligands, PD-L1 or PD-L2, expressed by both immune and tumour cells [28]. In addition, activated T cells can express an inducible co-stimulator (iCOS), a surface receptor that is structurally and functionally similar to CD28 and enhances expression of Th2-related interleukin (IL)-10 rather than immune activating IL-2 [29]. This immunosuppressive helper T cell phenotype can be further induced by the zinc-finger transcription factor GATA3, which regulates Th2 cytokine expression [30]. Infiltrating CD8^+^ T cells in GBM have been shown to bear an exhausted phenotype, with upregulated expression of PD-1, LAG-3, TIGIT and CD39 [31,32,33].

GBM is considered an immunologically “cold tumour” due to the immunosuppressive microenvironment [33,34] and few immunogenic tumour-associated antigens (TAAs) [35]. The rapidly growing tumour alters the balance of interaction between cancer and immune cells, by outstripping its metabolic resources and shifting to glycolysis [36]. Cancer cells also recruit and alter nearby stromal cells to aid the tumour cells in avoiding immune detection and destruction [37,38]. Competition for glucose results in low pH, which creates a hostile tumour microenvironment [39,40,41] and drives a local increase in immunosuppressive stromal cells (Figure 1). These cells secrete immune inhibitory growth factors and cytokines, including vascular endothelial growth factor (VEGF), which is primarily produced by microglia, myeloid-derived suppressor cells (MDSCs) and tumour-associated macrophages (TAMs). TAMs and regulatory T cells such as the CD8^+^ CD28^-^ FoxP3^+^ T_reg_ cells also secrete immunosuppressive IL-10 [33] which downregulates interferon gamma (IFNγ) and MHC I through the JAK/STAT pathway [42]. Accumulating MDSCs are rendered in an immature state [43], secreting IL-13, which when bound to IL13Rα2, send anti-inflammatory signals to promote the alternative M2 phenotype in macrophages, causing them to lose their cytotoxic function and instead become immunosuppressive. Tumour cells aid in this immunosuppression by secreting transforming growth factor β (TGFβ) and expressing both membrane-bound and soluble PD-L1, inducing exhaustion in T and NK cells [44,45]. Moreover, the molecular heterogeneity of GBM, observed as cells with wildly dissimilar molecular genetic aberrations within the same tumour mass [16,17] also allows the tumour microenvironment to evolve and adapt to the immunological selection pressure. These conditions lead to immune escape and an immune-edited microenvironment that is refractory to killing by cytotoxic lymphocytes, while promoting resistance to standard treatments. New, personalised therapies that specifically address these molecular, genetic and immunological challenges are sorely needed.

## 3. Past Experiences and Thinking forward with CAR T Cell Therapy

Strategies that harness the immune system to combat cancer using immunotherapy represent the fourth pillar in cancer treatment [46,47]. One of the major advances came from the development of the synthetic CAR, representing modified TCRs, that contain both an antigen-binding site and a T cell costimulatory domain that allows for autonomous, MHC-independent activation of the T cells [48,49]. They consist of an extracellular single-chain variable fragment (ScFv) derived from an antibody, and an intracellular domain with signalling motifs. Over time, adjustments made to CARs have targeted their structure (Figure 2); where first-generation CARs contained the intracellular CD3ζ from a TCR, second-generation CARs included the costimulatory CD28 or 4-1BB endodomains fused to CD3ζ (reviewed in [50]). Both CD28 and 4-1BB results in activation of the T cell via the transcription factor NFκβ, CD28 mainly through the PI3K pathway [51], and 4-1BB through TNF signalling pathways [52].

While the 4-1BB domain has been shown to improve persistence and expansion potential in CD8+ central memory T cells, CD28 was associated with more rapid tumour elimination, increased IL-2 secretion and preferential expansion of effector memory T cell subsets [51,53]. Third generation CARs were developed by combining both CD28 and 4-1BB domains connected to CD3ζ (Figure 2A), resulting in superior expansion and persistence compared to second generation CARs with only CD28 [54]. Finally, fourth generation CARs, also known as T cell redirected for universal cytokine-mediated killing (TRUCK) [55] or armoured CAR T cells [56], feature structures similar to second-generation CARs with the additional ability to induce expression of cytokine genes such as IL-12. Other CAR T constructs have been designed with the aim to improve their efficiency and reduce adverse effects. For instance, the tandem CAR (TanCAR), built for bispecific activation of T cells, mitigates antigen escape by targeting two antigens simultaneously [57]. Another alternative is using an ON-switch CAR, where activation relies on the administration of a pro-drug, which limits off-tumour toxicities by requiring dimerization via a priming small molecule [58], or Lenalidomide [59] (Figure 2B). The latter can also be used to operate an OFF-switch degradable CAR system with the same pro-drug using a different configuration of the CAR to cause ubiquitination and subsequent degradation of the construct.

To engineer CAR T cells, cancer patients must first undergo apheresis, and their T cells are isolated from whole blood. The T cells are then transduced or transfected, either through viral vectors or electroporation, resulting in the expression of a CAR designed to target a specific cell surface antigen. After selection, CAR-expressing T cells are expanded and cryopreserved, before reinfusion into the patient [60]. The patient is then monitored for efficacy and potential toxicity.

The first major success for human CAR T cell immunotherapy came with CARs that were designed to target CD19, a marker of differentiated B cells, in haematological malignancies [61]. Clinical trials reported complete remission in 90% of leukaemia patients one month after infusion of the modified T cells [61,62]. The tremendous success of this immunotherapy strategy triggered the approval of CAR T cells as the first gene-modified cell therapy by the U.S. Food and Drug Administration (FDA). CD19-CAR T cells were approved for the treatment of B cell lymphomas (axicabtagene ciloleucel-Yescarta^®^) [63] and leukaemia (tisagenlecleucel-Kymriah^®^) [64] in 2017. Since then, several iterations of the paradigm are under investigation in clinical trials.

The remarkable response in haematological malignancies stimulated interest in the research and development of new CARs for application in the treatment of solid tumours, including GBM. Several target antigens have been proposed for the application of CAR T cell therapy against GBM, some of which have been recently tested in clinical trials. Proof of concept has been provided by a case report showing transient complete response in one patient with advanced GBM treated with autologous engineered CAR T cells targeting IL13Rα2 [65]. Designer T cell persistence has been studied extensively, primarily in haematological malignancies, and recent research has shown a few avenues for improving this aspect of treatment. These include disrupting the methylcytosine dioxygenase *TET2* gene, resulting in improved persistence and expansion of CD19-directed CAR T cells in vivo [66], or targeting hematopoietic progenitor kinase 1 (HPK1), either by knockout of the gene or by specific proteolysis [67]. However, obstacles persist for effective, sustained CAR T cell therapy for solid cancers (reviewed in [68]). These include non-ubiquitous expression of the target antigen, physical barriers such as the blood–brain barrier (BBB) and extracellular matrix, immunosuppressive cells and cytokines, which are all present within a hostile, often hypoxic tumour microenvironment [34]. These obstacles are compounded by antigen loss [69] and serious adverse events observed in CAR T clinical trials, such as immune effector cell-associated neurotoxicity syndrome (ICANS) [70] and cytokine release syndrome (CRS) [71] in a brain encapsulated in an unyielding, bony cranium.

## 4. Challenging Solid Brain Tumours with Natural Killers

Another approach recently translated into clinical trials is CAR NK therapy. Originally, the only difference between CAR T and CAR NK cells was the lymphocyte that held the CAR. The CARs had the same structure in both lymphocytes, since CARs with the CD3ζ chain alone or in combination with other costimulatory domains (Figure 2) also promoted NK cell activation [72]. However, as the field advanced, domains from adaptor molecules associated with NK receptors such as DAP10 and DAP12 have been introduced. While CAR NK cells with a DAP10 domain showed poor efficiency [73], CAR NK cells with DAP12 demonstrated enhanced cytotoxicity when compared with those with CD3ζ [74].

Cytotoxic T cells and NK cells differ in many ways. Unlike T cells, NK cells are innate lymphocytes that specialise in recognising and killing transformed- and virus-infected cells through germline-encoded receptors that yield activating or inhibiting signals when interacting with target cell surface ligands [75,76,77]. Activating receptors include the natural cytotoxicity receptors NKp30 [78], NKp44 [79] and NKp46 [80], NK group 2 D (NKG2D) receptor [81], and killer immunoglobulin-like receptors (KIRs) with short cytoplasmic domains [82,83,84]. With these receptors, educated, cytotoxic CD56^dim^ NK cells can become activated and kill tumour cells through a variety of mechanisms [85,86]. We have shown that NK cells that selectively express particular KIRs potently kill GBM cells and GBM stem-like cells [84,87,88]. A recent study showed enhanced killing capability of NK cells against GBM stem-like cells when inhibiting αv integrin or TGFβ [89]. Unlike CAR T cells, in the event of antigen loss by the tumour cells, CAR NK cells may still possess the ability to kill by their intrinsic activating receptors [90,91,92,93,94]. In addition, CAR NK cells are considered safer than CAR T cells since they are not known to induce ICANS or CRS [95,96,97].

Importantly, allogeneic T cells are known to induce graft-versus-host disease (GvHD) while NK cells do not [95]. Autologous T cell therapy may not be amenable for all patients since those with significant lymphopenia due to prior anti-cancer treatments or those with a rapidly progressive disease will not be eligible due to insufficient numbers of T cells to expand into therapeutic doses of CAR T cells. This is important for GBM, as both treated and treatment-naïve GBM patients experience lymphopenia [98]. The protracted time required for CAR T cell production is also a bottleneck [97]. The first CAR NK cell approaches focused on primary peripheral blood-derived NK cells transfected with the CAR construct [99], but NK cells are more difficult to transfect than T cells [100]. To circumvent difficulties in introducing gene edits to NK cells, a recent publication has outlined a method for efficient electroporation of CRISPR-associated Cas9 ribonuclear protein (RNP) complexes in primary NK cells [101]. A similar protocol has been used to efficiently knock out the immune checkpoint receptor T-cell immunoglobulin mucin family member 3 (*TIM3*) in primary NK cells from healthy donors [102], PD-1 [103], and CD38 [104]. However, NK cells only represent approximately 5–15% of circulating lymphocytes in healthy adult humans [105] and require extensive ex vivo expansion. Additionally, they have a much shorter life span than T cells [106], although the persistence of CD-19-directed, IL-15 expressing CAR NK cells from cord blood were shown to survive for at least 4 weeks in mouse models [107]. In humans, multiple or higher doses of engineered NK cells might be required for a sustained treatment effect. NK cells may be obtained from various tissue sources, however, such as umbilical cord blood, induced pluripotent stem cells (iPSCs) or cell lines, such as NK-92 [108]. This characteristic provides the opportunity for “off-the-shelf” cells that can be applied to any patient, which is less readily achieved with autologous T cell therapies [100]. However, the possibility of a host-versus-graft scenario, where the implanted cells are systematically destroyed by the host immune system, must also be considered. Research into the production of reliable, “off-the-shelf” alternatives for cancer immunotherapy is currently in progress for both T and NK cells (reviewed in [109,110]). Strategies to avoid GvHD include knockout of the TCR in engineered T cells [111,112], while host attacks and graft rejection may be mitigated through knockout of MHC or application of lymphodepleting chemotherapeutics prior to engineered T/NK cell administration [113,114,115,116]. In GBM, NK cells have been shown to comprise only a small portion of immune cells in the tumour microenvironment, and to have a CD56^+^ CD16^-^ phenotype [33], indicating that these cells may not be capable of killing CAR-expressing cells. Another opportunity may lie in NK cells’ inherent expression of KIR2DL4, which can bind HLA G to inhibit host NK cells [117,118].

The heterogeneity of GBM creates a significant challenge for the identification of a ubiquitously expressed tumour specific target for CAR constructs. In addition, GBM cells grown in stem cell media, which most closely preserve the original tumour phenotype, tend to exhibit reduced expression of both classical and non-classical MHC class I ligands for inhibitory NK receptors [33,119]. Therefore, these conditions, if present within patient tumour microenvironments, may provide additional opportunities for receptor mediated cytotoxicity in CAR NK cells [120,121] via both KIR and natural cytotoxicity receptors against stress-induced ligands known to be expressed by GBM tumour cells [87,88,122]. Bispecific TanCAR NK cells could be used to mitigate possible tumour escape by targeting two antigens [57]. As such, the choice of either CAR T or NK cell therapy depends on the strategy utilised, as the two provide different advantages and disadvantages, as summarised in Table 1. Taken together, CAR NK cells represent potentially superior effectors for solid tumours such as GBM.

### In Vitro and In Vivo Clinical Studies for Designer NK Cells in GBM

The preclinical demonstration of the anti-tumour efficacy of CAR NK cells in haematological malignancies [73,124] has stimulated a number of clinical trials involving patients with these cancers (reviewed in [125]). Some CAR NK cell approaches proposed for the treatment of GBM have also demonstrated anti-tumour efficacy and extended animal survival in preclinical settings. Murakami et al. established a CAR KHYG-1 NK cell line which exhibited capacity for killing epidermal growth factor receptor (EGFR) variant III (EGFRvIII)-expressing GBM cells in vitro [126]. However, it lacked in vivo anti-tumour efficacy in mice ectopically injected with GBM cells [127]. Genßler et al. and Han et al. successfully generated CAR NK-92 cells bispecific for wild type EGFR and EGFRvIII and demonstrated their efficacy in GBM both in vitro and in vivo, when injected intracranially in GBM-bearing mice [128,129]. Furthermore, Zhang et al. showed the efficacy of another NK-92 CAR cell line, the NK-92/5.28.z, which is specific for human epidermal growth factor receptor 2 (HER2)/ErbB2, against GBM both in vitro and in vivo, through intratumoural injections into the brain [130]. Interestingly, these are the only CAR NK cell therapy constructs being tested in clinical trials for human GBM (clinical trial ID NCT03383978, CAR2BRAIN), although other avenues for CAR NK in GBM are in development (reviewed in [131]). The CAR2BRAIN study is in phase I, aiming to be completed by December 2022, with the primary goals of assessing safety and tolerability, persistence of NK-95/5.28.z cells and cytokine profile in cerebrospinal fluid and blood for 24 weeks after injection. The patients undergoing this trial receive only one injection of the engineered cells, and so far, no dose-limiting effects have been reported. Nevertheless, it remains to be seen if this and future trials will have to incorporate more doses to see a sustained benefit.

## 5. Picking a Suitable Target for GBM Immunotherapy

In GBM, heterogenous mutations are typically not immunogenic [15,132], which renders the tumours resistant to new generation immunotherapies, including CAR T and immune checkpoint blockade (reviewed in [133,134]). Choosing one or more suitable targets is critical for a sustained, beneficial effect. In addition to absent or limited expression in healthy tissues, the target antigen must have persistent expression on the tumour cells and present an appropriate affinity to the CAR which allows recognition of the target with low off-target toxicity [135]. An overview of immunotherapeutic targets for the treatment of GBM currently in clinical trials is shown in Table 2, which includes EGFRvIII, HER2/ErbB2, interleukin-13 receptor alpha 2 (IL13Rα2), B7H3 and NKG2D ligands. Although T cell exhaustion may be limited by targeting CTLA-4 and PD-1, chemotherapeutics with monoclonal antibodies against CTLA-4, PD-1 and PD-L1 have shown mixed and unpredictable response in GBM [136]. In a recent publication, however, Cloughesy et al. performed a randomised, early phase clinical trial using neoadjuvant pembrolizumab, a PD-1 inhibitor, which showed both overall and progression-free survival benefit to patients with recurrent GBM when applied both before and after surgery [137]. These results emphasize the importance of timing of treatment.

### 5.1. Growth Factor Receptors as Targets

One of the initial targets for GBM immunotherapy trials with CAR T cells was EGFRvIII, a constitutively active mutated variant of the EGFR receptor. Currently, there are ten active CAR T treatments for GBM in clinical trials, two of which target EGFRvIII (clinical trial ID NCT01454596 and NCT03726515). Results from different studies are contradicting based on the independent prognostic value of EGFRvIII in GBM for overall survival of newly diagnosed patients with EGFRvIII positive tumours [141,142]. A phase I clinical trial reported that a single infusion of EGFRvIII-directed CAR-T cell led to loss of antigen, and induction of adaptive resistance through increased PD-L1, IDO1, IL-10 and the presence of FoxP3^+^ T_reg_ cells [143]. These results highlight the problems of early antigen loss and swift immunosuppressive response in these patients. HER2/ErbB2 is structurally homologous to EGFR, and mRNA has been found in 76% of primary GBM cell lines [144]. However, its expression levels in normal brain tissue, especially neuronal and endothelial cells of the cerebral cortex [145], pose difficulties in developing immunotherapy options with tolerable side-effects for brain tumour application. Nevertheless, pre-clinical studies demonstrated HER2-specific T lymphocyte cytotoxicity against GBM stem-like cells that was not evident in HER2^-^ cells [139]. In addition, HER2-directed CAR NK cells engineered from healthy donors and breast cancer patients have been shown to selectively kill HER2^+^ tumour cells while avoiding healthy cells in vitro [146]. HER2 is also currently the only target used for CAR NK cell therapy in human trials for GBM with the CAR2BRAIN study in Germany (clinical trial ID NCT03383978), as mentioned above.

### 5.2. Interleukin-13 Receptor Alpha 2

IL13Rα2 is another cancer-associated receptor currently used in CAR T cell therapy for GBM (clinical trial ID NCT04003649 and NCT02208362). Overexpression in GBM coupled with low to no detectable levels in healthy brain tissue render IL13Rα2 a suitable target for immunotherapy [138]. The native IL13Rα2 has been reported to induce an invasive phenotype in GBM by promoting epithelial-mesenchymal transition. However, through interaction with EGFRvIII, IL13Rα2 signalling induces proliferation through activation of the STAT3 pathway [147]. Thus, dual targeting of IL13Rα2 and EGFRvIII could be a promising strategy for GBM CAR therapy. One study has reported on the administration of IL13Rα2 CAR T cells intracranially post resection, followed by further infusions both intraventricular and intrathecally in a single patient [65]. Although the patient experienced disease recurrence after adoptive cell therapy, remarkable tumour shrinkage was observed in all lesions by 77–100%, exhibiting a sustained responses for 7.5 months after intrathecal administration.

### 5.3. Co-Stimulator B7H3

B7H3 (CD276) is a costimulatory protein on the cell surface, where the B7 superfamily of proteins was initially thought to be solely involved in immune co-stimulation [148,149]. It has since been found to be highly expressed on GBM and tumour-associated endothelial cells, with protein levels increasing with increasing tumour grade [140,150]. Although its functional role is not fully elucidated, B7H3 is associated with poorer prognosis in GBM patients [150], making it a potential target for therapy. A recent clinical report using CAR-T cells directed towards B7H3 showed in one patient with recurrent GBM who received seven infusions significantly reduced volume of contrast enhancing lesion on MRI and clinical response for 50 days [151]. Two B7H3-directed CAR T cell clinical trials for GBM are currently in recruitment in China (clinical trial ID NCT04077866 and NCT04385173).

### 5.4. Multitarget Approach with NKG2D

The TAAs mentioned above are all considered feasible targets for GBM immunotherapy. However, their heterogenous expression on cancer and normal cells present challenges with toxicity, antigen escape and subsequent recurrence. One suggested approach to overcome this issue has been to use the lectin-like, type 2 transmembrane receptor NKG2D in CAR therapy. NKG2D is an activating receptor expressed on the surface of NK and CD8^+^ T cells [33,152,153] that binds MHC class-related chains A and B (MICA and MICB, respectively), as well as UL16-binding protein 6 (ULBP6), that are typically only expressed during hyper-proliferation and transformation [154]. NKG2D ligands are expressed by stressed and transformed cells, potentially induced as a response to chemo- and radiotherapy [81,155]. In immunocompetent mice, a recent study reported that an NKG2D-based CAR T therapy worked synergistically with radiotherapy, yielding persistence and long-term protection from disease [156]. Weiss et al. incorporated the full-length NKG2D protein fused to CD3ζ in association with DAP10, functioning similarly to a second-generation CAR, and transduced it into T cells. These promising results provide evidence for clinical translation into human trials. However, the single approved NKG2D-based CAR T clinical trial (clinical trial ID NCT04270461) has since been withdrawn due to administrative reasons.

## 6. Genetic Modification as an Aid to Existing Therapies

Cutting edge gene editing technologies [157] provide the opportunity to enhance the efficacy of existing molecular immunotherapy strategies. Introduction of clustered regularly interspaced short palindromic repeats (CRISPR) and the CRISPR-associated endonuclease Cas (typically Cas9) to design T or NK cells may improve the potency of cancer immunotherapy. CRISPR/Cas9 technology is used to knockout genes by inducing double-stranded DNA breaks (DSB) at specific sites within the genome. The Cas9 enzyme forms a complex with a 20-base guide RNA (gRNA) that is predesigned to recognize a complementary DNA target site in a gene of interest. Cas9 locates a protospacer adjacent motif (PAM) sequence directly downstream of the gRNA sequence, binds and triggers a DSB at the genomic target site [158] (Figure 3). The cell then repairs the damage through mechanisms based on the cell cycle stage and proliferative status [159,160]. The most common repair pathway is non-homologous end joining, wherein the two ends of the DNA are fused, often resulting in faulty repair, with random insertion or deletion of nucleotides causing frameshift mutations and thus a knockout of the gene. The other, less common pathway, homology-directed repair (HDR), uses a DNA repair template to repair the damaged DNA, or introduces a gene edit, as with HDR CRISPR [161]. HDR CRISPR, however, is difficult to accomplish and often leads to high numbers of unwanted insertions and deletions (indels) outside the genetic area of interest, so-called off-target effects [162,163].

The use of CRISPR/Cas9 technology in the development of cancer therapy has only recently begun to have an impact on human trials, as the ethical use of CRISPR/Cas9 to edit the human genome has been a hotly debated topic [164,165]. CRISPR/Cas9 mutagenesis screening has been used as a tool to identify REGNASE-1 as a negative regulator of antitumour responses, whose knockout improved efficiency and persistence of effector T cells in vitro [166]. In addition, CRISPR/Cas9 was used to successfully generate *CBLB* knockout NK cells derived from human placental stem-cells, without altering their phenotype [167], indicating possibility for use in primary lymphocytes. Preclinical research has successfully used multiplex CRISPR/Cas9 to limit universal CAR T cell exhaustion by disrupting the *PD-1* gene, as well as the endogenous TCR α chain (*TRAC*) and β-2 microglobulin (*B2M*) [112]. The T cells, directed against EGFRvIII, were administered intracerebrally in GBM bearing mice, and enhanced survival. In a recent phase I human clinical trial, CRISPR/Cas9 was used on autologous T cells to knock out the endogenous TCR α and β chains as well as *PD-1* in 3 patients with refractory cancer: 2 patients with multiple myeloma, and 1 with myxoid and round cell liposarcoma [111]. In addition, a transgenic TCR was transduced into the cells, targeting the cancer testis antigen NY-ESO-1. The T cells were then heterogeneously expanded and intravenously reinfused into the patients. The engineered cells persisted in the patients for up to 9 months. Stadtmauer et al. showed that at the time of infusion, the amount of Cas9 protein had diminished to a non-detectable level in the engineered T cells, had acceptable levels of off-target editing, and that the patients did not develop humoral responses to Cas9. This pilot trial indicates that CRISPR/Cas9 is feasible in combination with cancer immunotherapy and could increase persistence of adoptively transferred cells in patients, by limiting exhaustion through knockdown of *PD-1* in the T cell graft [111]. Although the cancers in this trial were not of CNS origin, similar strategies could be applied to T cells from GBM patients.

Although CRISPR/Cas9 technology has been rapidly improved and adopted broadly in biology, non-specific gene modifications and mutations do arise at off-target sites [168], especially when targeting multiple genes simultaneously. A novel, more elegant CRISPR-derived method, prime editing, has recently been developed, which has resulted in higher specificity and exact nucleotide alterations for CRISPR/Cas9 gene editing [162]. Prime editing uses a combination of a Cas9 nickase (H840A) with reverse transcriptase (RT), an RNA-dependent DNA polymerase, as well as an elongated gRNA sequence called the prime editing guide RNA, to find and replace selected nucleotides. The flexibility and specificity of prime editing could be used to augment existing therapies, for example, to engineer deletions of up to 80 base pairs, transversions, transitions, or insertions of up to 44 base pairs, and has been successfully used in patient-derived disease models [169]. Prime editing could be an ideal method to use in tandem with existing immunotherapies such as CAR T and NK cell therapy in GBM. This technology could be used, for example, to induce specific mutations in regulatory genes to enhance effector cell efficiency by promoting expression of proinflammatory cytokines such as *IL-12*, or to prolong their persistence by mutating exhaustion markers *CTLA-4* or *PD-1*, while avoiding unwanted translocations and indels. The benefit of using this technology as an alternative to traditional CRISPR/Cas9 may be the ability to generate engineered cell populations with predictable genetic alterations and to expand homogenous subsets only with the desired edits.

## 7. Experimental Targets for GBM Therapy

As our understanding of the evolving, heterogenous GBM microenvironment improves, so do the repertoire of strategies available for discovering and testing novel therapeutic targets. Currently, pre-clinical data shows promise for several TAAs (Table 3), which also includes further research into NKG2D as mentioned above.

### 7.1. Synergistic Effects Targeting Chondroitin Sulfate Proteoglycan 4 (CSPG4)

A promising GBM TAA is chondroitin sulfate proteoglycan 4 (CSPG4), also known as neural glial antigen 2 (NG2), a transmembrane signalling proteoglycan primarily involved in cell migration and angiogenesis [174,175,176]. CSPG4/NG2 is overexpressed in approximately 58% of GBM patient biopsies and is independently prognostic for poorer patient outcome [177]. CSPG4/NG2 has been found to contribute to chemo- and radioresistance in preclinical studies [177,178]. Although expression within the tumour microenvironment is heterogenous, CSPG4/NG2 was inducible by TNFα secreted by activated, resident microglia [170]. Preclinical research has indicated that CSPG4-directed CAR T cells combined with concomitant TNFα may be an amenable strategy for GBM treatment. However, TNF signalling in the brain is tightly regulated, and promoting TNF levels could lead to neuronal toxicity and adverse neurological effects (reviewed in [179]).

### 7.2. The Pan-Population Antigen MR1

A recent paper by Crowther et al. used genome-wide CRISPR-Cas9 screening to find an HLA independent target for TCRs. They reported that the monomorphic MHC class 1-related protein (MR1) led to the specific killing of malignant cells, while leaving noncancerous cells intact [171]. This result suggests the existence of a pan-population TAA that is cancer-specific, which thus provides an excellent molecular target for the development of future clinical trials, for both solid tumours and haematological cancers.

### 7.3. Targeting Extracellular ATP

Another potential new target is CD39 (ectonucleoside triphosphate diphosphohydrolase 1), an ectonucleotidase that binds extracellular ATP, converting it to extracellular adenosine in cooperation with CD73, and mediating immune suppression [180,181]. Extracellular ATP is associated with dead or dying cells, and is a hallmark of immunogenic cell death (ICD) [182] and a pro-inflammatory tumour microenvironment. CD39 is upregulated in multiple cancers, including GBM [33,183], as a response to hypoxia and tissue damage, and is expressed by tumour cells, MDSCs and TAMs. Several immune cell subsets including NK cells and CD4^+^CD28^+^ T_reg_ [184] express CD39 where it suppresses NK cytotoxicity [185]_._ Recently, a study by Simoni et al. reported that the presence of CD39 on CD8^+^ T cells indicate exhaustion, and represent a subset of tumour-infiltrating T cells that are not specific for tumour antigens [186]. Pre-clinical data from syngeneic and patient derived xenograft models demonstrated that treatment with monoclonal anti-CD39 enhanced T-cell proliferation and Th1 cytokine production, facilitated infiltration of T cells into tumours and rescued anti-PD-1 resistance [172]. Thus, two phase 1 clinical trials for solid tumours are currently investigating anti-CD39 antibodies in monotherapy or combination therapies (clinical trial ID NCT03884556 and NCT04261075).

### 7.4. Clinical Value of 2-HG in IDH-Mutant Gliomas

The *IDH1* mutation has been shown to confer an immunologically quiescent phenotype, with fewer tumour infiltrating lymphocytes (TILs) and reduced expression of PD-L1 in mutant compared to IDH-WT tumours [187]. Bunse et al. showed that 2-HG secreted by the tumour cells is taken up by T cells, where it suppresses proliferation and activation by inhibiting TCR signalling [173]. Their results support clinical investigation of a combination therapy of IDH inhibitors and immunotherapy, such as adoptive T cell transfer or immune checkpoint blockade (reviewed in [188]). While clinical trials for high grade gliomas mainly focuses on IDH-WT patients, this strategy would aid in the development of treatments for IDH-mutant gliomas which are also ultimately terminal for patients. Immunotherapy trials directed towards IDH-mutant gliomas are underway, however, with recent results from the NOA16 vaccine trial showing immune responses in more than 90% of study participants with newly diagnosed IDH1 mutant glioma [189].

## 8. Developing Combination Approaches

Using a combination of living and non-living drugs has the advantage of targeting multiple pathways and requiring lower dosages, limiting the chance of drug resistance. Drug resistance presents one of the greatest challenges in the treatment of GBM, as TMZ resistance is seen in GBM after recurrence, even when MGMT is absent in the original tumour [190]. To make combination therapies more applicable in GBM, the synergistic or antagonistic properties of individual drugs must be scrutinised, as the genetic heterogeneity may result in unexpected consequences. TMZ resistance in GBM was investigated with mathematical models, which revealed changes in gene expression patterns in GBM that were associated with irreversible TMZ resistance [191]. Combination therapies that target TAAs within the same or intertwining pathways may thus help to increase treatment efficacy, as the tumour adapts to avoid destruction. An example of this strategy for GBM is to use CAR T cell therapy against both CD39 and CD73 simultaneously, either with a TanCAR or two separate CARs, to halt the degradation of extracellular ATP by targeting the rate-limiting aspects of the pathway (Figure 4). Anti-tumour effects have been shown in vitro and in vivo when both these targets were inhibited simultaneously with monoclonal antibodies [192]. With an altered approach, a combination of CRISPR/Cas9 knockout of *PD-1* from patient T cells to improve persistence [111], and viral transduction to express two CARs targeted against both CD39 and CD73, could result in potent anti-cancer effect by increasing immunogenicity of the microenvironment. To safeguard against serious adverse events as a result of T cell over-activation, an inducible Caspase 9 can also be introduced into the cells, which has been shown to rapidly and specifically eliminate modified T cells [193]. Enhanced expression of CD47, a “don’t eat me” signal, could also be used to avoid destruction by TAMs [194].

For future combination trials, timing of the administration of immunotherapy with CAR T or NK cells must be considered in relation to standard therapy, to ensure a persistent anti-tumour effect without killing the engineered cells with chemo- or radiotherapy. While chemotherapy causes ICD of tumour cells [195], which could activate existing and transferred T cells through the release of tumour neoantigens, TMZ and radiotherapy are lymphotoxic and induce lymphopenia [196,197]. Combination trials with designer T or NK cell approaches could in this case be administered as adjuvant therapy, following standard therapy, to take advantage of the newly available neoantigens, while avoiding chemotherapy- and radiation-induced toxicity, thus optimising synergy. Alternatively, neoadjuvant administration of anti-PD-1 in recurrent GBM has shown survival benefit, with increased expression of IFNγ-related genes [137]. In addition, recent research has demonstrated the combined use of 4-1BB agonism with anti-PD-1 therapy to combat TIL exhaustion in GBM [198]. Thus, treating patients prior to surgery with anti-PD1 antibody with or without 4-1BB agonism, followed by adjuvant CAR T cells with a third generation CD28/4-1BB-based CAR construct, could take advantage of the less “cold” tumour microenvironment to eliminate and control peripheral disease. With this strategy, prime editing could be used to develop homogenous, multiplex genome-edited T cells with reduced expression of *PD-1* and endogenous TCR.

## 9. Mode of Delivery—Anti-Tumour Effect vs. Adverse Effects

New cancer therapy designs must include suggestions on how best to deliver the treatment to the patient for maximum, sustained benefit. For GBM, treatment delivery is complicated by the presence of the BBB, a specialized barrier composed of endothelial cells held together by tight junctions [199]. The BBB restricts passage of cells [200], and passive entry of hydrophobic compounds and large molecules >100 kDa from systemic circulation or interstitial fluids into the brain parenchyma [201], protecting the diffusely infiltrating tumour cells [202]. The BBB imposes limitations on drug delivery by preventing transport of most small-molecule drugs and nearly 100% of larger therapeutics into the brain parenchyma [203]. Even when penetration is achieved, drug therapy results in high intracranial pressure to maintain an effective pharmacological concentration through diffusion from blood to the CNS [204]. However, the BBB is variably disrupted in GBMs [205,206], as corroborated by the frequently observed peritumoural oedema [207], indicating feasibility of immunotherapy to the brain. The disrupted BBB in GBM is further supported by the presence of tertiary lymphoid structures in preclinical glioma models, associated with increased T cell infiltration in the tumour microenvironment [208]. Studies demonstrated that although T cell infiltration varied between patients [209,210,211], high infiltration correlated with longer survival [211,212,213,214], in GBM patients [33]. These findings are consistent with an immunoactive environment that may benefit from local delivery of immune modulatory therapies.

Local delivery options for live drugs include administration by intraventricular or intrathecal methods. Intrathecal delivery is promising with regard to targeting specific areas of the brain, although not all drugs are acceptable intrathecal candidates, as they can cause inflammation of the meninges and overall high drug concentrations in the CSF [215,216]. For GBM, intraventricular dosing of chemotherapy was studied, but this mechanism provides little exposure of the parenchyma to the drug, resulting in poor distribution [217]. However, following intracavity administration, intraventricular delivery of IL13Rα2-directed CAR T cells was performed to improve trafficking to multifocal tumour lesions through the CSF, with remarkable efficiency in one patient [65]. These results indicate that for recurrent GBM patients with distal lesions, intrathecal administration with CAR T or NK cells could provide sufficient distribution.

Finally, once sufficient pharmacological drug concentrations are achieved, the next step is to keep the treatment localized without causing excessive death of normal cells or graft rejection. Any new treatment against a molecular target expressed in non-malignant healthy cells must be delivered optimally to avoid such toxicities, as observed in some clinical cases [218]. For GBM, this is especially critical when considering systemic delivery, as many TAAs used in clinical trials and experimental models are also expressed outside the brain, such as HER2 [219].

## 10. Conclusions and Perspectives

GBM is an aggressive, infiltrative cancer with poor prognosis that requires new outside-the-box strategies for effective treatment, particularly in the case of recurrence. Designer CAR T and NK cell therapies are steadily improving for solid cancers, and with gene editing technologies such as CRISPR/Cas9 having entered human trials, new possibilities emerge that have the potential to improve efficacy, while catering to the characteristics of individual patients’ tumours and minimizing adverse events. We have seamlessly highlighted current strategies to improve anti-cancer targeting and suggested novel prospective combinations that may be undertaken to advance the field with a focus on NK and T cells. NK cells are particularly attractive effector cells because they are present at early stages of inflammation and release cytokines that modulate the microenvironment with receptor-ligand interactions that cross talk with several cell types. Macrophages and microglia comprise a major subset of immune cells in GBM [33], however, their evolving phenotypic plasticity results in great diversity and redundancy. This might present a moving target that would pose challenges to choosing the right populations to manipulate using CRIPSR/Cas9 technology. Future directions should aim to develop this strategy also for these cell types.

## Figures and Tables

**Figure 1 cancers-13-04986-f001:**
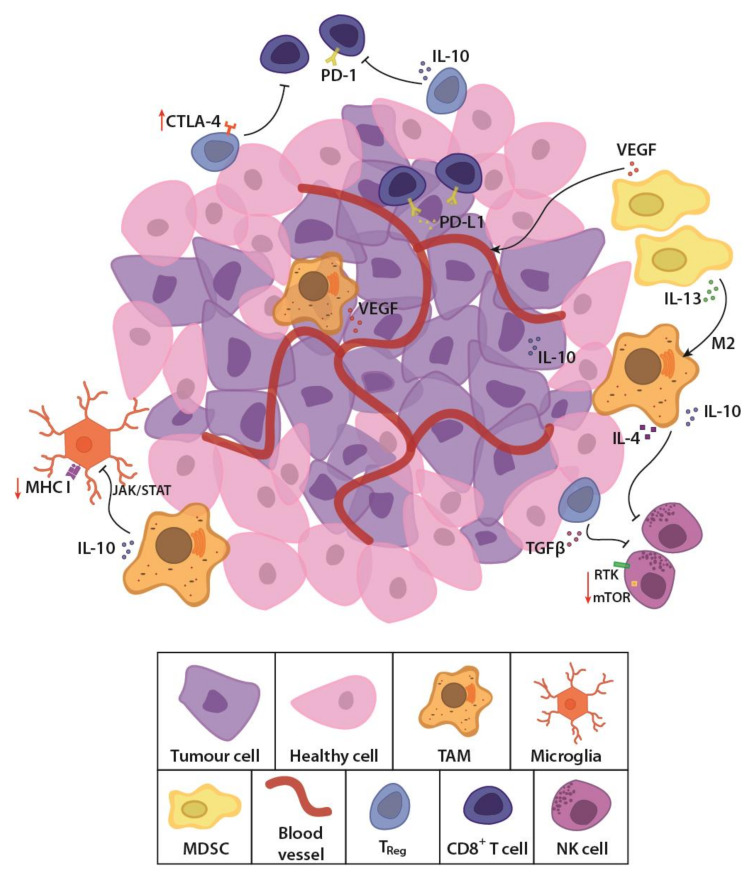
Schematic representation of the glioblastoma (GBM) tumour microenvironment. The evolving tumour microenvironment consists of diffusely infiltrative tumour cells with a leaky neovasculature, anti-inflammatory stromal cells and high concentrations of immunosuppressive cytokines and signalling molecules. Together, these factors generate a hostile environment where proliferation and persistence of cytotoxic immune cells are significantly hampered. TAM, tumour-associated macrophage; MDSC, myeloid derived suppressor cell; T_reg_, regulatory T cell; NK cell, natural killer cell; TGFβ, transforming growth factor β; PD-1, programmed cell death protein 1; PD-L1, programmed death-ligand 1; CTLA-4, cytotoxic T lymphocyte-associated protein-4; IL-13, interleukin-13; IL-10, interleukin-10; VEGF, vascular endothelial growth factor; MHC I, major histocompatibility complex class I; JAK/STAT, Janus kinase/signal transducers and activators of transcription; RTK, receptor tyrosine kinase; mTOR, mammalian target of rapamycin.

**Figure 2 cancers-13-04986-f002:**
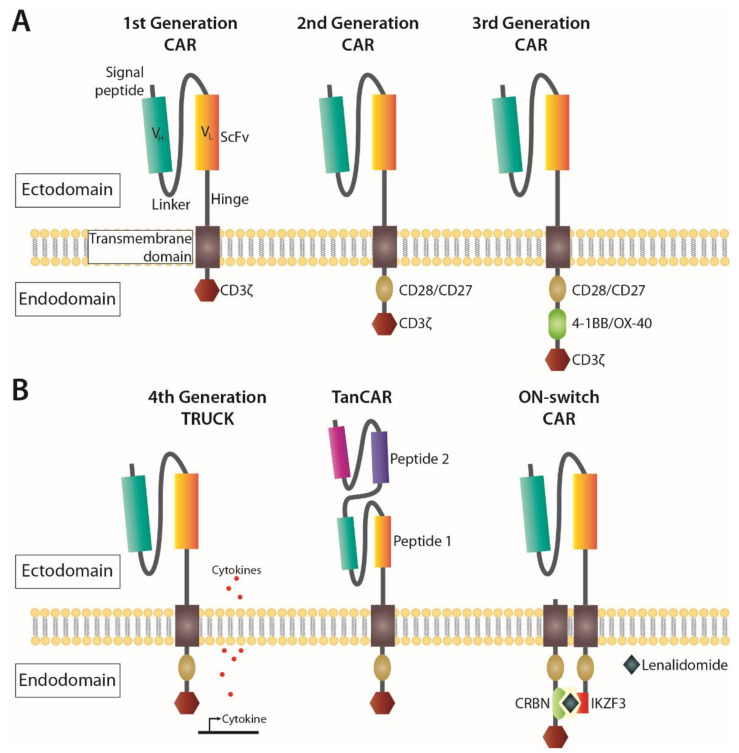
Schematic representation of chimeric antigen receptor (CAR) constructs. The ectodomain of a CAR contains a single-chain variable fragment (ScFv) region, which is derived from the antigen-binding fragment (Fab) of a monoclonal antibody containing a heavy-chain (V_H_) linked to a light-chain (V_L_), all linked to the transmembrane domain by a hinge region. The ScFv region contains an antigen recognition domain, which is specific for the desired tumour antigen to be targeted. (**A**) The first three generations of CAR constructs made alterations to the endodomain, which contains the intracellular signalling domain of the zeta (ζ) chain of the T cell receptor (TCR)/CD3 complex, and co-stimulatory signalling domains (CD27, CD28, 4-1BB or OX-40). (**B**) Later iterations of CARs based on the 2^nd^ generation CAR endodomain included induction of activating cytokines (4^th^ generation TRUCK), antigen recognition domains for two antigens (TanCAR) or required the addition of a small molecule, or Lenalidomide, for the dimerization (through zinc finger domains CRBN and IKZF3) and activation of the CAR (ON-switch CAR).

**Figure 3 cancers-13-04986-f003:**
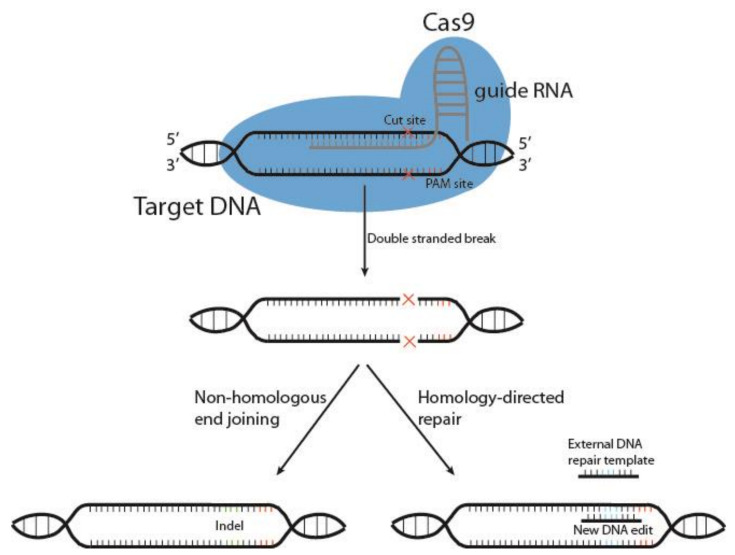
Schematic representation of CRISPR/Cas9 gene editing. The DNA endonuclease Cas9 is introduced to target cells via transduction, transfection or addition of pure Cas9 protein, and complexes with the guide RNA (gRNA). This complex between Cas9 and gRNA searches for a protospacer adjacent motif (PAM) sequence in proximity to the gRNA 20-base sequence. Once bound, Cas9 causes a double-stranded DNA break, and the cell must use either non-homologous end joining or homology-directed repair to fix the break. This results in either wild type DNA, edited DNA, or insertions and deletions (indels) that result in gene knockout.

**Figure 4 cancers-13-04986-f004:**
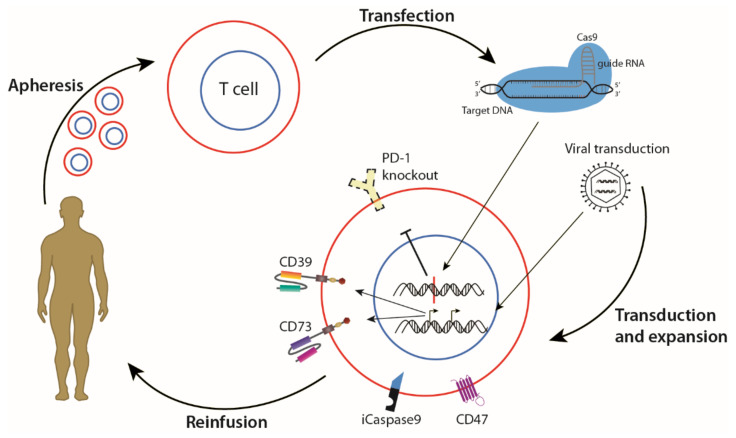
Combination CAR T cell therapy with gene editing to treat GBM. T cells are isolated from the blood of a cancer patient. CRISPR/Cas9 gene editing technology can then be used to knock out inhibitory receptor PD-1, and viral transduction induces expression of chimeric antigen receptors (CARs) targeting tumour-associated antigens CD73 and CD39. To avoid serious adverse events, inducible Caspase 9 (iCaspase9) can be activated to swiftly eliminate the adoptively transferred cells. The altered T cells can then be expanded and re-introduced to the cancer patient. Image adapted with permission from [111] from AAAS. gRNA, guide RNA; PD-1, programmed cell death protein 1; CD39, cluster of differentiation (CD) 39 (Ectonucleoside triphosphate diphosphohydrolase-1); CD73, (5′-nucleotidase); CD47, (integrin-associated protein).

**Table 1 cancers-13-04986-t001:** Comparisons between CAR T and NK cell therapies.

Comparisons to Consider	CAR T	CAR NK
Cell source	Autologous setting	Allogeneic setting or cell lines [108]
Off-the-shelf	Not eligible	Eligible [100]
GvHD	Induces GvHD	Not known to induce GvHD
Gene delivery	Effective by transfection or transduction	Effective by Cas9 RNP
Patients as candidates	Restricted to patient’s condition	All patients are potential candidates
Off-target effects	ICANS/CRS [70,71]	No/minimal ICANS or CRS
Response to antigen loss	No alternative mechanism	Response by existing NK cell receptors
Infusion dose	Single/few doses	Multiple or high
In vivo persistency	Longer (several months) [123]	Shorter (days to weeks) [106]
FDA/EMA approval	Approved [63,64]	Not yet approved
Cancer indication	Haematological malignancies [61]	Alternative for solid tumours

GvHD, graft versus host disease; RNP, ribonuclear protein; ICANS, immune effector cell-associated neurotoxicity syndrome; CRS, cytokine release syndrome; FDA, U.S. Food and Drug Administration; EMA, European Medicines Agency.

**Table 2 cancers-13-04986-t002:** Current targets used in CAR T and NK cell therapy for GBM.

Clinical Trial ID	Target	Cell Type/Expression	Trial Method	Trial Status	Reference
NCT03726515, NCT01454596	EGFRvIII	Glioma cells, 64% of GBM cases	CAR T	Completed, completed	[138]
NCT03383978, NCT01109095	HER2/ErbB2	Tumour cells, normal endothelial and neuronal brain cells (low expression)	CAR NK	Recruiting, completed	[139]
NCT04003649, NCT02208362	IL13Rα2	Tumour cells, TAMs, MDSCs, overexpressed in 50% of GBM cases	CAR T	Recruiting, recruiting	[138]
NCT04077866, NCT04385173	B7H3	DC, T cells, overexpression on GBM tumour cells	CAR T	Recruiting, recruiting	[140]
NCT04270461	NKG2DL	Receptor expressed on NK, NKT and T cells, ligands expressed on tumour cells	CAR T	Withdrawn	[90]

EGFRvIII, epidermal growth factor receptor variant III; HER2, human epidermal growth factor receptor 2; ErbB2, Erb-B2 Receptor Tyrosine Kinase 2; IL13Rα2, interleukin 13 receptor subunit alpha 2; B7H3, B7 homologue 3; NKG2D, natural killer group 2 member D; TAM, tumour-associated macrophage; MDSC, myeloid-derived suppressor cell; DC, dendritic cell; NKT, natural killer T cell.

**Table 3 cancers-13-04986-t003:** Experimental targets for future GBM cancer therapy.

Potential New Targets	Significance of Target	Reference
CSPG4	Pre-clinical data suggests combination CAR T therapy with TNFα treatment for GBM.	[170]
MR1	A pan-population MHC class 1-related protein. T cells with TCRs targeting MR1 potently kill cancer cells, and not healthy cells.	[171]
NKG2D	Multiple ligands, less likely to lead to immune escape.	[156]
CD39	Promotes T cell proliferation and Th1 cytokine release.	[172]
2-HG	In IDH-mutant gliomas, IDH inhibition combined with immunotherapy might enhance T cell activity.	[173]

CSPG4, chondroitin sulfate proteoglycan 4; MR1, major histocompatibility complex class 1-related protein; NKG2D, natural killer group 2 member D; CD39, cluster of differentiation 39 (ectonucleoside triphosphate diphosphohydrolase-1); 2-HG, 2-hydroxyglutarate.

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
