# Peer review of "Challenges and Prospects for Designer T and NK Cells in Glioblastoma Immunotherapy"

_cancers, 2021, doi:10.3390/cancers13194986_

Round 1

Reviewer 1 Report

The authors summarized cutting-edge CAR T and NK therapy against GBM and perspective them. Although the paper is quite well written and clearly presented, several modifications are needed.

Major revision

Title: “Gene editing and molecular immunology in glioblastoma: Opportunities and challenges for effective treatment” 

Please consider the title change.

e.g. “Challenges and Perspective for Designer T and NK cells in glioblastoma therapy.”

Abstract:

“Molecular immunotherapy” would not be considered to suitable term since included broad meaning scope. Surely, this term was used in the reference “Front. Oncol., 09 2020 | https://doi.org/10.3389/fonc.2020.569017”. This paper has many aspects of immunology including cytokine, tumor vaccine, pattern recognition receptors, antibody, immune checkpoint inhibitor, cell therapy, and so on.

I consider that “Designer T and NK cell-based immunotherapy” is suited instead of “molecular immunology”.

To my knowledge, “Designer cell” is used by Culliton BJ for the first time (see reference: Fighting cancer with designer cells. Culliton BJ Science, 01 Jun 1989, 244(4911):1430-1433 DOI: 10.1126/science.2660264 1978.). “Designer cell” is revisited and genome editing and CAR introducing T and NK cells often called designer T and NK cell.

Main text:

1. In this review, the authors did not mention that genome editing enhances long-term persistency and effector functions in CAR T and NK cells. As it is called armored CAR T or NK cell. For example, TET2 deficiency enhances CAR T cell persistence human in vivo. At the experimental level, Regnase and HPK1 enhance CAR T function and persistency in vivo. Since there are likely to be other references, you must summarize together with the following references.

・Disruption of TET2 promotes the therapeutic efficacy of CD19-targeted T cells Nature. 2018 Jun;558(7709):307-312. doi: 10.1038/s41586-018-0178-z

・Targeting REGNASE-1 programs long-lived effector T cells for cancer therapy. Nature. 2019 Dec;576(7787):471-476. doi: 10.1038/s41586-019-1821-z.

・Hematopoietic Progenitor Kinase1 (HPK1) Mediates T Cell Dysfunction and Is a Druggable Target for T Cell-Based Immunotherapies. Cancer Cell. 2020 Oct 12;38(4):551-566.e11. doi: 10.1016/j.ccell.2020.08.001.

2. Now, CRISPR/Cas9 edited NK cell is attracting attention. Because it is easy that guide RNA/Cas9 protein (RNP) is electroporated directly into primary NK cells. Representative references are listed below. There would be other references about genome-edited NK cells. Please summarize them. This information indicated that armored CAR NK could provide more anti-tumour benefits.

・CISH-edited NK: Drug target validation in primary human natural killer cells using CRISPR RNP. J Leukoc Biol. 2020 108(4):1397-1408. doi: 10.1002/JLB.2MA0620-074R.

・TGFBR2: Targeting the αv integrin/TGF-β axis improves natural killer cell function against glioblastoma stem cells. J Clin Invest. 2021 Jul 15;131(14):e142116. doi: 10.1172/JCI142116.

・CD38-edited NK: CD38 knockout natural killer cells expressing an affinity optimized CD38 chimeric antigen receptor successfully target acute myeloid leukemia with reduced effector cell fratricide. Haematologica. 2020 Dec 30;Online ahead of print. doi: 10.3324/haematol.2020.271908.

・CBLB-edited NK: CBLB ablation with CRISPR/Cas9 enhances cytotoxicity of human placental stem cell-derived NK cells for cancer immunotherapy. J Immunother Cancer. 2021 Mar;9(3):e001975. doi: 10.1136/jitc-2020-001975.

・PD-1-edited NK: Development of a protein-based system for transient epigenetic repression of immune checkpoint molecule and enhancement of antitumour activity of natural killer cells. Br J Cancer. 2020 Mar;122(6):823-834. doi: 10.1038/s41416-019-0708-y.

・TIM-3 edited NK: CRISPR-Cas9-Mediated TIM3 Knockout in Human Natural Killer Cells Enhances Growth Inhibitory Effects on Human Glioma Cells. Int J Mol Sci. 2021 Mar 28;22(7):3489. doi: 10.3390/ijms22073489.

Minor revision

1. Line 35-36: “The incidence of brain and central nervous system (CNS) tumours was 5.7/100,000 in Europe in 2020 [1]”

Reference 1 is described worldwide data, not European data.

2. Line 101-104: In addition, regulatory T cells such as the CD8+ CD28-FoxP3+ Treg cells inhibit cytotoxic T and NK cells through secretion of cytokines such as transforming growth factor β (TGFβ), tumour necrosis factor α (TNFα), and soluble and membrane-bound PD-L1 [31].

Is Reference 31 described IL-10 and CTLA-4, not TGF, TNF, and PD-L1? And, does TNFα inhibit cytotoxic T and NK cells' function?

3. Line 192-194: “Proof of concept has been provided by a case report showing transient complete response in one patient with advanced GBM treated with autologous engineered CAR T cells [61]”

TheA specific ligand for CAR must be described. (IL-13Rα)

4. Line 217-218: killer immunoglobulin-like receptors (KIRs) with short cytoplasmic domains [76-78]

Reference 76 was described p40. Is P40 KIR?

5. Line 251-252: Another opportunity may lie in NK cells’ inherent expression of KIR2DL4, which can bind HLA-G to inhibit host NK cells [105-107].

Is Reference 107 described KIR or HLA-G? I could not confirm it.

6. Line 266:

Table 1 is hard to understand. I suggest to change the following table. This does not necessarily mean that you should do as in the table below.

CAR T

CAR NK

Cell source

Autologous setting

Allogeneic setting or cell lines

Off the shelf

Not eligible

Eligible

GvHD

+

+/- [96] (acute GvHD occur?)

Gene delivery

Effective

Resistance

Patients candidate

Restricted patient’s conditions

Potentially all patients

Off-target effect

(ICANS/CRS) [64,65]

No/minimal ICANS or CRS

The ability to respond to antigen loss

No alternative mechanism

Respond by NK cell receptors

Injection dose

A single/few doses

Multiple or high

In vivo persistency

longer (** to ** mounts)

shorter (** to ** mounts)[95]

FDA/EMA approval

Approval [59,60]

Not yet approval

Indication

haematological malignancies [57]

Alternative for Solid tumours

7. Line 303-304: An overview of immunotherapeutic targets for the treatment of GBM currently in clinical trials is shown in Table 2.

Please insert the word of specific target EGFRvIII, HER2/ErbB2, IL13Rα2, B7H3, NKG2DL. It would be easy to understand for readers.

8. Line 354-362

Clinical trials for GBM using B7H3-directed CAR T cell is published below. Please add the following reference.

Administration of B7-H3 targeted chimeric antigen receptor-T cells induce regression of glioblastoma. Xin Tang, Yuelong Wang, Jianhan Huang, Zongliang Zhang, Fujun Liu, Jianguo Xu, Gang Guo, Wei Wang, Aiping Tong & Liangxue Zhou. Signal Transduction and Targeted Therapy volume 6, Article number: 125 (2021)

9. Line 428,

Add reference [99] at the end of the sentence.

10. Line 339, 354, 363, 471, 478, 495

Please insert a new line above subheadings.

11. Line 517-522

Is this sentence your original strategy? Readers would be hard to distinguish the reported strategy or new one.

12. Line 553

This is out of my scope, but don't you need to submit the permit document?

13. Line 609-610:Organisation, W.H. GLOBOCAN 2020, Cancer Incidence and Mortality Worldwide: IARC CancerBase. Available online: 610 https://gco.iarc.fr/today/data/factsheets/populations/900-world-fact-sheets.pdf (accessed on 21/07).1.

I do not think that reference number 1 is the correct description?

14. Line 630-631: H. Blakstad, J.B., M.A. Rahman, V.S. Arnesen, P. Brandal, S.A. Lie, M. Chekenya, D. Goplen. Survival in a consecutive series 630 of 467 recurrent glioblastoma patients: impact of prognostic factors and treatment at two independent institutions 2021

Submitting the manuscript is not suitable for reference. Should be added to the reference list by registered with preprints.

15. Line 995

TNFa is changed to TNFα

Author Response

We thank this reviewer for their unbelievably dedicated effort, constructive and brilliant suggestions that definitely improved the quality of our manuscript.  Please find below detailed point-by point account of the changes made.

Reviewer rebuttal

Reviewer 1

Comment nr 1: Title: “Gene editing and molecular immunology in glioblastoma: Opportunities and challenges for effective treatment” 

Please consider the title change.

e.g. “Challenges and Perspective for Designer T and NK cells in glioblastoma therapy.”

Abstract:

“Molecular immunotherapy” would not be considered to suitable term since included broad meaning scope. Surely, this term was used in the reference “Front. Oncol., 09 2020 | https://doi.org/10.3389/fonc.2020.569017”. This paper has many aspects of immunology including cytokine, tumor vaccine, pattern recognition receptors, antibody, immune checkpoint inhibitor, cell therapy, and so on.

I consider that “Designer T and NK cell-based immunotherapy” is suited instead of “molecular immunology”.

To my knowledge, “Designer cell” is used by Culliton BJ for the first time (see reference: Fighting cancer with designer cells. Culliton BJ Science, 01 Jun 1989, 244(4911):1430-1433 DOI: 10.1126/science.2660264 1978.). “Designer cell” is revisited and genome editing and CAR introducing T and NK cells often called designer T and NK cell.

Author response: The authors agree that “designer” T and NK cells is a better suited term to be used in this review and have changed the title and contents to reflect this, incorporating the reference to Culliton BJ 1989.

Comment nr 2: In this review, the authors did not mention that genome editing enhances long-term persistency and effector functions in CAR T and NK cells. As it is called armored CAR T or NK cell. For example, TET2 deficiency enhances CAR T cell persistence human in vivo. At the experimental level, Regnase and HPK1 enhance CAR T function and persistency in vivo. Since there are likely to be other references, you must summarize together with the following references.

・Disruption of TET2 promotes the therapeutic efficacy of CD19-targeted T cells Nature. 2018 Jun;558(7709):307-312. doi: 10.1038/s41586-018-0178-z

・Targeting REGNASE-1 programs long-lived effector T cells for cancer therapy. Nature. 2019 Dec;576(7787):471-476. doi: 10.1038/s41586-019-1821-z.

・Hematopoietic Progenitor Kinase1 (HPK1) Mediates T Cell Dysfunction and Is a Druggable Target for T Cell-Based Immunotherapies. Cancer Cell. 2020 Oct 12;38(4):551-566.e11. doi: 10.1016/j.ccell.2020.08.001.

Author response: The authors agree that further mention of research into enhancing persistence and efficacy of CAR T and NK cells was required. The suggested references have been included in the main text.

Comment nr 3: Now, CRISPR/Cas9 edited NK cell is attracting attention. Because it is easy that guide RNA/Cas9 protein (RNP) is electroporated directly into primary NK cells. Representative references are listed below. There would be other references about genome-edited NK cells. Please summarize them. This information indicated that armored CAR NK could provide more anti-tumour benefits.

・CISH-edited NK: Drug target validation in primary human natural killer cells using CRISPR RNP. J Leukoc Biol. 2020 108(4):1397-1408. doi: 10.1002/JLB.2MA0620-074R.

・TGFBR2: Targeting the αv integrin/TGF-β axis improves natural killer cell function against glioblastoma stem cells. J Clin Invest. 2021 Jul 15;131(14):e142116. doi: 10.1172/JCI142116.

・CD38-edited NK: CD38 knockout natural killer cells expressing an affinity optimized CD38 chimeric antigen receptor successfully target acute myeloid leukemia with reduced effector cell fratricide. Haematologica. 2020 Dec 30;Online ahead of print. doi: 10.3324/haematol.2020.271908.

・CBLB-edited NK: CBLB ablation with CRISPR/Cas9 enhances cytotoxicity of human placental stem cell-derived NK cells for cancer immunotherapy. J Immunother Cancer. 2021 Mar;9(3):e001975. doi: 10.1136/jitc-2020-001975.

・PD-1-edited NK: Development of a protein-based system for transient epigenetic repression of immune checkpoint molecule and enhancement of antitumour activity of natural killer cells. Br J Cancer. 2020 Mar;122(6):823-834. doi: 10.1038/s41416-019-0708-y.

・TIM-3 edited NK: CRISPR-Cas9-Mediated TIM3 Knockout in Human Natural Killer Cells Enhances Growth Inhibitory Effects on Human Glioma Cells. Int J Mol Sci. 2021 Mar 28;22(7):3489. doi: 10.3390/ijms22073489.

Author response: The authors agree that highlighting the use of Cas9 RNP technology and similar strategies to induce gene edits in NK cells is an important topic to include. The suggested references, in addition to other similar methods (Liu E. et al., Leukemia, 2018, doi: 10.1038/leu.2017.226).

Comment nr 4: Line 35-36: “The incidence of brain and central nervous system (CNS) tumours was 5.7/100,000 in Europe in 2020 [1]”

Reference 1 is described worldwide data, not European data.

Author response: Numbers and reference have been updated to the newly published European Cancer burden, Dyba T. et al., European Journal of Cancer, 2021, doi: 10.1016/j.ejca.2021.07.039.

Comment nr 5: Line 101-104: In addition, regulatory T cells such as the CD8+ CD28-FoxP3+ Treg cells inhibit cytotoxic T and NK cells through secretion of cytokines such as transforming growth factor β (TGFβ), tumour necrosis factor α (TNFα), and soluble and membrane-bound PD-L1 [31].

Is Reference 31 described IL-10 and CTLA-4, not TGF, TNF, and PD-L1? And, does TNFα inhibit cytotoxic T and NK cells' function?

Author response: References have been expanded to include TGFβ (Han J. et al., Am J Cancer Res, 2015, PMCID: PMC4449428) and PD-L1 (Hao C. et al., Front Oncol, 2020, doi: 10.3389/fonc.2020.01015). TNFα has been shown to induce cytotoxicity in NK and T cells, and no supporting references were found indicating that it has an inhibitory function. Figure 1 has been updated to remove TNFα and the main text has been altered.

Comment nr 6: Line 192-194: “Proof of concept has been provided by a case report showing transient complete response in one patient with advanced GBM treated with autologous engineered CAR T cells [61]”

TheA specific ligand for CAR must be described. (IL-13Rα)

Author response: The CAR specific ligand has been added here.

Comment nr 7: Line 217-218: killer immunoglobulin-like receptors (KIRs) with short cytoplasmic domains [76-78]

Reference 76 was described p40. Is P40 KIR?

Author response: p40 is a regulator of lymphocyte activation and proliferation, but it is not a KIR. The reference has been removed.

Comment nr 8: Line 251-252: Another opportunity may lie in NK cells’ inherent expression of KIR2DL4, which can bind HLA-G to inhibit host NK cells [105-107].

Is Reference 107 described KIR or HLA-G? I could not confirm it.

Author response: The reference describes LAK cells, not NK cells specifically. LAK cells comprise both NK and T cell phenotypes, however the authors elected to remove the reference for clarity.

Comment nr 9: Line 266:

Table 1 is hard to understand. I suggest to change the following table. This does not necessarily mean that you should do as in the table below.

Author response: Thank you for the good suggestion, it was hard to improve upon the suggested table format.

Comment nr 10: Line 303-304: An overview of immunotherapeutic targets for the treatment of GBM currently in clinical trials is shown in Table 2.

Please insert the word of specific target EGFRvIII, HER2/ErbB2, IL13Rα2, B7H3, NKG2DL. It would be easy to understand for readers.

Author response: Target names have been added to the main text.

Comment nr 11: Line 354-362

Clinical trials for GBM using B7H3-directed CAR T cell is published below. Please add the following reference.

Administration of B7-H3 targeted chimeric antigen receptor-T cells induce regression of glioblastoma. Xin Tang, Yuelong Wang, Jianhan Huang, Zongliang Zhang, Fujun Liu, Jianguo Xu, Gang Guo, Wei Wang, Aiping Tong & Liangxue Zhou. Signal Transduction and Targeted Therapy volume 6, Article number: 125 (2021)

Author response: The mention and reference have been added to the appropriate paragraph.

Comment nr 12: Line 428,

Add reference [99] at the end of the sentence.

Author response: Reference has been added.

Comment nr 13: Line 339, 354, 363, 471, 478, 495

Please insert a new line above subheadings.

Author response: Lines have been added above all subheadings.

Comment nr 14: Line 517-522

Is this sentence your original strategy? Readers would be hard to distinguish the reported strategy or new one.

Author response: Monoclonal antibodies have previously been used in combination against CD39 and CD73 simultaneously, Perrot I., et al., Cell reports, 2019, doi: 10.1016/J.CELREP.2019.04.091. The suggestion of combining a dual targeted approach using CARs against CD39 and CD73, in addition to CRISPR/Cas9 knockout of PD-1 with inducible Caspase 9 and/or CD47 is our original suggestion for future combination trials.

Comment nr 15: Line 553

This is out of my scope, but don't you need to submit the permit document?

Author response: Permit document should now be accessible. In addition, specific permission from the American association for the advancement of science (AAAS) has been added to figure legend.The journal offers gold open access, which allows for modification as long as permission is granted, and the original work is cited.

Comment nr 16: Line 609-610:Organisation, W.H. GLOBOCAN 2020, Cancer Incidence and Mortality Worldwide: IARC CancerBase. Available online: 610 https://gco.iarc.fr/today/data/factsheets/populations/900-world-fact-sheets.pdf (accessed on 21/07).1.

I do not think that reference number 1 is the correct description?

Author response: Thank you. The authors have investigated the correct citation method for referring to GLOBOCAN fact sheets and have updated reference 1 accordingly.

Comment nr 17: Line 630-631: H. Blakstad, J.B., M.A. Rahman, V.S. Arnesen, P. Brandal, S.A. Lie, M. Chekenya, D. Goplen. Survival in a consecutive series 630 of 467 recurrent glioblastoma patients: impact of prognostic factors and treatment at two independent institutions 2021

Submitting the manuscript is not suitable for reference. Should be added to the reference list by registered with preprints.

Author response: According to the Cancers guidelines to authors, unpublished works may be referenced in a manner they suggest. We have adapted the reference accordingly.

Comment nr 18: Line 995

TNFa is changed to TNFα

Author response: Thank you. The alteration has been made.

Reviewer 2 Report

The authors extensively described the literature regarding CAR-T and CAT-NK  in cancer immune therapies. Although the limited research regarding glioblastoma immunotherapy using gene-editing technology, the authors summarized the CAR-T- and CAT-NK-related studies.  Given the crucial role of glioblastoma-associated macrophages in the tumor microenvironment, macrophage-based cell therapies in cancer treatment should be described in the review. Additionally, it is better to summarize what future directions of glioblastoma immunotherapy based on genome editing are for readers in the last section of Conclusions and perspectives. 

Author Response

We are grateful to the review for their constructive comments to our work

Reviewer 2

Comment nr 1: The authors extensively described the literature regarding CAR-T and CAT-NK in cancer immune therapies. Although the limited research regarding glioblastoma immunotherapy using gene-editing technology, the authors summarized the CAR-T- and CAT-NK-related studies. Given the crucial role of glioblastoma-associated macrophages in the tumor microenvironment, macrophage-based cell therapies in cancer treatment should be described in the review.

Author response: This review focuses entirely on the use of T and NK based therapies for the treatment of GBM. In particular, NK cells are attractive effector cells due to 3 primary reasons. Firstly, NK cells are present at early stages of the inflammatory process. Secondly, they secrete pro-inflammatory cytokines that can shape the tumour microenvironment, including macrophages, indicating that they are feasible effector cells for immune responses. Thirdly, NK cells have multiple receptors capable of interacting with many cell types, many of which can be used to create designer molecular immunotherapy. We agree that the use of macrophages is an interesting direction in the field. However, given the phenotypic diversity and redundancy within the tumour associated macrophages and microenvironment induced phenotypic plasticity may present a moving target, that would pose challenges to choosing the right populations to manipulate during the evolving course of the disease using CRISPR/Cas9 technology. To the best of our knowledge, there are no studies using this technology to specifically address this topic. Given these reasons, this topic is considered beyond the scope of this current work, but could be considered in a different review.

Comment nr 2: Additionally, it is better to summarize what future directions of glioblastoma immunotherapy based on genome editing are for readers in the last section of Conclusions and perspectives. 

Author response: As indicated by the new title, all future prospects on designer T and NK cells are addressed seamlessly throughout the main text. Future perspectives have been altered to reflect this and are addressed in conclusions, line 633-643.

Round 2

Reviewer 1 Report

The authors have adequately addressed my concerns.